# Differential Analysis of Venous Sinus Diameters: Unveiling Vascular Alterations in Patients with Multiple Sclerosis

**DOI:** 10.3390/diagnostics14161760

**Published:** 2024-08-13

**Authors:** Abdulkadir Tunç, Gurkan Danisan, Onur Taydas, Ahmet Burak Kara, Samet Öncel, Mustafa Özdemir

**Affiliations:** 1Department of Neurology, Faculty of Medicine, Sakarya University, 54100 Sakarya, Turkey; 2Department of Radiology, Faculty of Medicine, Sakarya University, 54100 Sakarya, Turkey; gurkandanisan@yahoo.com (G.D.); taydasonur@gmail.com (O.T.); drmstfrd@gmail.com (M.Ö.); 3Department of Neurology, Sakarya University Training and Research Hospital, 54100 Sakarya, Turkey; sametoncel@yahoo.com

**Keywords:** multiple sclerosis, cerebral veins, magnetic resonance imaging, neuroimaging, disease progression, vascular pathophysiology

## Abstract

Background: Alterations in the cerebral venous system have been increasingly recognized as a significant component of the pathophysiology of multiple sclerosis (MS). This study aimed to explore the relationship between venous sinus diameter and MS to understand potential vascular alterations in MS patients compared with controls. We sought to determine whether these alterations were correlated with disease characteristics such as duration, lesion type, and disability score. Methods: This study included 79 MS patients diagnosed according to the 2017 McDonald criteria and 67 healthy individuals. Magnetic resonance imaging (MRI) scans via a 1.5 Tesla system provided measurements of the superior sagittal sinus, right and left transverse sinus, sinus rectus, and venous structures. Statistical analysis was conducted via SPSS, employing independent sample t tests, ANOVA, chi-square tests, and Pearson correlation analysis, with the significance level set at *p* < 0.05. Results: This study revealed significant differences in venous sinus diameter between MS patients and controls, with MS patients exhibiting larger diameters. Specifically, patients with brainstem and spinal lesions had larger diameters in certain sinus regions. No significant correlations were found between venous sinus diameter and demographic factors, expanded disability status scale scores, or lesion counts. However, a significant increase in perivenular lesions was noted in patients with longer disease durations. Conclusions: The findings indicate notable vascular alterations in MS patients, particularly in venous sinus diameters, suggesting a potential vascular component in MS pathology. The lack of correlation with conventional clinical and MRI metrics highlights the complexity of MS pathology. These insights underscore the need for further research, particularly longitudinal studies, to elucidate the role of venous changes in MS progression and their potential as therapeutic targets.

## 1. Introduction

Multiple sclerosis (MS) is a chronic, immune-mediated disorder that impacts the central nervous system (CNS), leading to a diverse array of neurological symptoms. Its etiology and pathophysiology are complex and involve genetic, environmental, and immunological factors [1,2,3]. Recently, the focus has shifted toward understanding the role of vascular changes, particularly in the cerebral venous system, in MS progression. The involvement of the venous system in MS is well documented, with the central vein sign being a typical pathognomonic magnetic resonance imaging (MRI) feature that reflects the perivenular localization of typical MS lesions [4,5,6]. Vascular abnormalities affecting cerebrospinal fluid (CSF) circulation may play a significant role in MS pathophysiology. Studies suggest that MS progression is linked to “subpial surface-in” and “ependymal-in” gradients, causing leptomeningeal and cortical damage, as well as deep gray matter damage [7,8]. Perivascular inflammation is predominant in MS and has been associated with early demyelinating activity and disease progression [9]. In a study comparing the hemodynamics of clinically isolated syndrome (CIS), relapsing-remitting MS (RRMS), and control patients, there was a 71% increase in cerebral blood volume (CBV) in normal-appearing white matter and a 59% increase in deep gray matter in the CIS. This increase in CBV in the CIS was attributed to either arteriolar dilatation due to inflammation or neovascularity [10]. Additionally, there is evidence of a significant increase in bridging vein transmural pressure in MS patients, with MS patients exhibiting larger cortical veins correlated with significant fatigue [11]. The venous sinus alterations observed in MS patients may be a consequence of intrathecal inflammation, involving the accumulation of lymphoid-like infiltrates, B lymphocytes, and other immune cells. Large parenchymal veins are found close to the ependymal surface and drain toward the periventricular surface. Perivascular spaces have also been shown to increase in size and number in the MS brain. Therefore, abnormalities of the venous system in MS are likely an epiphenomenon of intrathecal inflammation rather than a primary cause of the disease [7,9].

Additionally, other hypotheses have been proposed to explain the vascular alterations observed in MS. The cerebral venous system, which is crucial for maintaining cerebral homeostasis, may undergo alterations in venous sinus diameters associated with various neurological conditions. These alterations are thought to reflect the interplay between CSF circulation, intracranial pressure, and cerebral blood flow. In MS, such changes could signal physiological aberrations, impacting CSF absorption and venous outflow, akin to those observed in normal pressure hydrocephalus [12,13]. The concept of pulse wave encephalopathy, which is traditionally linked to Alzheimer’s disease, vascular dementia, and normal pressure hydrocephalus, is also relevant in MS and encompasses the effects of pulse waves generated in the craniospinal cavity on the venous system [1]. In MS, the observed alterations in arterial stroke volume and venous system compliance suggest that changes in the Windkessel effect could direct more pulsation energy into the venous outflow, thereby affecting CSF dynamics and contributing to MS pathophysiology [12]. Reductions in venous outflow compliance in MS, defined as the change in volume divided by the change in pressure, may indicate alterations in venous pressure and impact cerebral fluid dynamics [13,14]. Sex-specific differences in MS incidence and progression are well documented, with a higher incidence in females but more aggressive progression in males. This highlights potential gender-based variations in MS pathology, including venous anatomy and dynamics [15,16].

This study focuses on venous sinus diameter, aiming to contribute to the discourse on the role of cerebral venous physiology in MS pathogenesis and progression. By evaluating these diameters in MS patients compared with a control group, we sought to identify significant differences and explore correlations with disease characteristics such as disease duration, lesion types, and disability scores. Understanding these relationships could pave the way for novel diagnostic and therapeutic approaches in MS management.

## 2. Materials and Methods

### 2.1. Study Population

This study included 79 patients with a definite diagnosis of MS, aged between 18 and 55 years, who were diagnosed according to the 2017 McDonald criteria [17]. These patients were regularly followed up at the MS outpatient clinic of our hospital. The control group consisted of 67 healthy individuals recruited from general neurology outpatient clinics. Initially, 100 patients were planned for each group. However, owing to insufficient MRI quality during the analysis process, the final sample comprised 79 patients with MS and 67 individuals in the control group. This study adhered to the ethical standards established by the Declaration of Helsinki. Prior to initiation, the study received approval from the institutional review board and ethics committee (Sakarya University Faculty of Medicine noninterventional clinical research ethics committee; approval number: 216). Informed consent was obtained from all individual participants included in the study. We ensured that our sampling was as random as possible within the constraints of our study population. Patients eligible for this study had not experienced MS exacerbation in the last three months and did not have accompanying vasculitis, diabetes, hypertension, or any underlying disease that might mimic MS lesions. Similarly, the control group participants did not have any known diseases affecting the CNS. Patients with cerebral venous anatomical variation in both the MS group and the control group were excluded from the study. For all participants, demographic data, disease duration, Expanded Disability Status Scale (EDSS) scores, disease-modifying agent use, and CSF parameters, including oligoclonal bands (OCB) and IgG indices and band numbers, were meticulously recorded.

### 2.2. MRI Protocol

MRI scans were performed using a 1.5 Tesla MRI system (GE Healthcare, Chicago, IL, USA). The imaging protocol included T1, T2, FLAIR, and susceptibility-weighted angiography (SWAN) sequences. T1-weighted imaging included the following parameters: repetition time (TR) = 500 ms, echo time (TE) = 12 ms, slice thickness = 5 mm, and field of view (FOV) = 240 mm; T2-weighted imaging with TR = 4000 ms, TE = 100 ms, slice thickness = 5 mm, and FOV = 240 mm; fluid-attenuated inversion recovery (FLAIR) with TR = 9000 ms, TE = 120 ms, inversion time (TI) = 2500 ms, slice thickness = 5 mm, and FOV = 240 mm; and susceptibility-weighted angiography (SWAN) with a multi-TE readout technique featuring TE1 = 7 ms, TE2 = 14 ms, TE3 = 21 ms, TR = 45 ms, flip angle = 15 degrees, slice thickness = 2 mm, and FOV = 240 mm. The SWAN sequence is particularly sensitive and implements a multiple echo time (multi-TE) readout technique, producing an enhanced signal-to-noise ratio and improved sensitivity information. A typical whole-brain acquisition with 3D, submillimeter resolution took approximately four minutes. In the SWAN sequence of MR images, the diameter of the superior sagittal sinus (SSS) from the vertex level, the right and left transverse sinuses, the sinus rectus, and the number of venous structures from the centrum semiovale level were quantitatively calculated manually. Measurements were made for each sinus on the basis of the widest diameter in the relevant region on axial images (Figure 1A–D). In addition, “central extinction” was assessed, defined as a reduction in signal intensity within the central venous structures on SWAN images, indicating potential iron accumulation or other susceptibility effects. This metric was quantitatively calculated to correlate with clinical and paraclinical features in MS patients. This analysis aimed to compare these measures in MS patients and healthy volunteers to ascertain whether there are anatomical changes in the brain venous system associated with MS. Additionally, the quantity and quality of iron accumulation in MS plaques revealed by the SWAN sequence were correlated with clinical and paraclinical features in MS patients. All the MRI findings were independently interpreted by two experienced neuroradiologists with a minimum of 5 years of expertise in the field. Discrepancies were resolved by consensus.

### 2.3. Statistical Analysis

The study involved the analysis of patient data via IBM’s statistical package for the social sciences (SPSS) for MacOS version 29.0 (IBM Corp., Armonk, NY, USA). For categorical data, frequencies and percentages were used, whereas means and standard deviations were employed for continuous data as descriptive statistics. The normality of variables was assessed via the Kolmogorov–Smirnov test. To compare two groups, the independent sample *t*-test was used, whereas ANOVA was used for comparisons involving more than two groups. The chi-square test was used to compare categorical variables. Additionally, Pearson correlation analysis was used to examine the relationships between continuous variables. The results were considered statistically significant if the *p*-value was less than 0.05.

## 3. Results

### Study Population and Demographics

Our study evaluated 146 participants, consisting of 79 (54.1%) patients with MS and 67 (45.9%) individuals in the control group. Among these, 116 (79.5%) were women, and 30 (20.5%) were men, with an average age of 36 years. The detailed clinical and demographic characteristics of the MS patient group are presented in Table 1. The average disease duration in the MS group was 7 years, with a majority (87.3%, 69 patients) having RRMS. The mean EDSS score was 2.2; brainstem lesions were present in 47.4% (37 patients), and spinal lesions were present in 80.8% (63 patients). A comparative analysis of variables between the patients with MS and healthy control subjects is shown in Table 2. A statistically significant difference was observed in age and all venous sinus diameters (SSS, sinus rectus, right transverse sinus, and left transverse sinus) between the groups (*p* < 0.05), with MS patients having greater values than the control group. To evaluate whether the differences in venous diameter between groups were influenced by age, we reanalyzed the venous diameter after adjusting for the effect of age. The results indicated that the statistically significant differences in venous diameter between the MS and control groups persisted even after controlling for age (Table 2).

The comparison of venous sinus diameters in the MS patient group, based on demographic and clinical findings, is outlined in Table 3. A statistically significant difference was noted in the SSS diameter and sinus rectus measurement in patients with spinal lesions (*p* < 0.05). Among MS patients, those with spinal lesions presented larger SSSs and sinus rectus diameters. Table 4 illustrates the relationships between venous sinus diameter and demographic and clinical findings in the MS patient group. An inverse relationship was observed between the age of the patients and the measurement of the right transverse sinus (*p* < 0.05). When patients were categorized according to EDSS score and disease duration, no significant relationships were found with venous sinus diameter, total lesion count, or the number of gadolinium-enhancing (gd+) lesions among all patients (Appendix A). However, the number of perivenular lesions was significantly greater in patients with a disease duration of more than 10 years (*p* = 0.032). No significant relationships were identified between sex, OCB positivity, the IgG index, the OCB count, central extinction, the brainstem functional score, or venous sinus diameter (*p* > 0.05) (Table 4 and Appendix A).

## 4. Discussion

Our findings highlight significant differences in venous sinus diameters between MS patients and a control group, corroborating the growing body of evidence suggesting vascular alterations in MS patients. The enlarged venous sinus diameters in MS patients, particularly in the SSS and sinus rectus, align with the concept of altered cerebral venous physiology in MS patients. These changes are likely a consequence of intrathecal inflammation, involving the accumulation of lymphoid-like infiltrates, B lymphocytes, and other immune cells, rather than simple adaptations to altered CSF dynamics and intracranial pressure. This is supported by evidence showing large parenchymal veins close to the ependymal surface and an increase in the size and number of perivascular spaces in the MS brain [7,9,18,19].

The observed increase in venous sinus diameter, especially in the presence of spinal lesions, suggests a possible association between venous pathology and MS lesion development. This finding is consistent with recent studies emphasizing the role of venous abnormalities in the pathophysiology of MS [18,20]. The increased venous sinus diameters in patients with spinal lesions could indicate a specific vascular pattern associated with certain MS lesion locations, furthering our understanding of MS heterogeneity. The hydraulic effectiveness of different segments of the sinuses, with variable shapes in cross-sections, was also considered. The sagittal and transverse sinuses tend to be triangular in the cross-section, whereas the sigmoid sinuses are more oval, impacting the efficiency of venous flow [21]. Interestingly, our results did not reveal a significant relationship between venous sinus diameter and demographic factors such as sex, which contrasts with the findings of some studies suggesting sex differences in MS pathology [22,23]. This discrepancy might be due to different study designs or population characteristics, and it warrants further investigation. Importantly, our study adjusted for the effect of age to ensure that the observed differences in venous sinus diameters were not biased by age differences between the groups. The re-evaluation indicated that the statistically significant differences in venous diameter between the MS and control groups persisted even after controlling for age. This finding reinforces the conclusion that the observed venous diameter differences are due primarily to MS pathophysiology rather than to age differences. The literature suggests that substantial venous diameter changes due to aging are typically observed over longer periods, such as decades. Huang et al. demonstrated that internal cerebral vein diameters increase significantly with age, particularly in older populations spanning 20–90 years [24]. Therefore, a four-year age difference is unlikely to account for the significant differences observed in our study, supporting the view that venous diameter differences are more likely attributable to the pathophysiological impacts of MS than to age alone [11].

The venous sinus alterations observed in our study could be a consequence of intrathecal inflammation, involving the accumulation of lymphoid-like infiltrates, B lymphocytes, and other immune cells [25,26,27]. This is further supported by evidence showing large parenchymal veins close to the ependymal surface and an increase in the size and number of perivascular spaces in the MS brain [7,9]. These venous system abnormalities are likely epiphenomenon of intrathecal inflammation rather than a primary cause of the disease [28,29]. The absence of significant correlations between venous sinus diameters and EDSS scores, as well as lesion counts, suggests that while venous changes are present in MS, they may not directly correlate with disease severity or lesion burden as measured by conventional clinical and MRI metrics. This finding aligns with that of Bateman et al., who reported complex relationships between venous pathology and MS clinical manifestations [14]. The significant increase in perivenular lesions in patients with longer disease durations could indicate progressive venous involvement over time in MS, supporting the hypothesis that venous pathology may play a role in the chronic progression of MS [18,30,31].

This study has several strengths, including a comprehensive dataset with a robust sample size, enhancing the reliability of the analysis. Detailed MRI analysis, including the use of advanced techniques such as the SWAN sequence, provides high accuracy and sensitivity in measuring venous sinus diameters. However, there are also limitations to consider. The cross-sectional design limits the ability to draw causal inferences about the relationship between venous sinus diameter and MS progression. This study was conducted at a single center, and the findings may not be generalizable to broader populations with different demographic or clinical characteristics. The lack of longitudinal data makes it challenging to determine how venous changes evolve over time and their impact on disease progression. Additionally, potential confounders, such as lifestyle factors and comorbidities, were not fully accounted for, which could influence venous sinus diameter. Importantly, although our study controlled for many variables, the age difference between groups should be considered when the results are interpreted. However, the literature suggests that this difference is unlikely to have a significant impact over such a short duration, a conclusion further supported by our adjusted analysis.

## 5. Conclusions

Our study provides significant evidence of vascular alterations in MS patients, demonstrating that MS patients exhibit notably larger venous sinus diameters, specifically in the SSS and sinus rectus, than healthy controls do. This finding supports the hypothesis that MS involves distinct changes in cerebral venous physiology. Notably, our additional analysis confirmed that these venous diameter differences are attributable to MS pathophysiology rather than minor age differences between groups, reinforcing the robustness of our results. These insights underscore the importance of considering venous pathology in MS research and clinical practice. Our study highlights the potential of venous assessments as biomarkers for MS activity and progression, paving the way for future research. Longitudinal and multicenter studies are essential to validate these findings and investigate the underlying mechanisms, potentially leading to novel therapeutic targets and improved management strategies for MS.

## Figures and Tables

**Figure 1 diagnostics-14-01760-f001:**
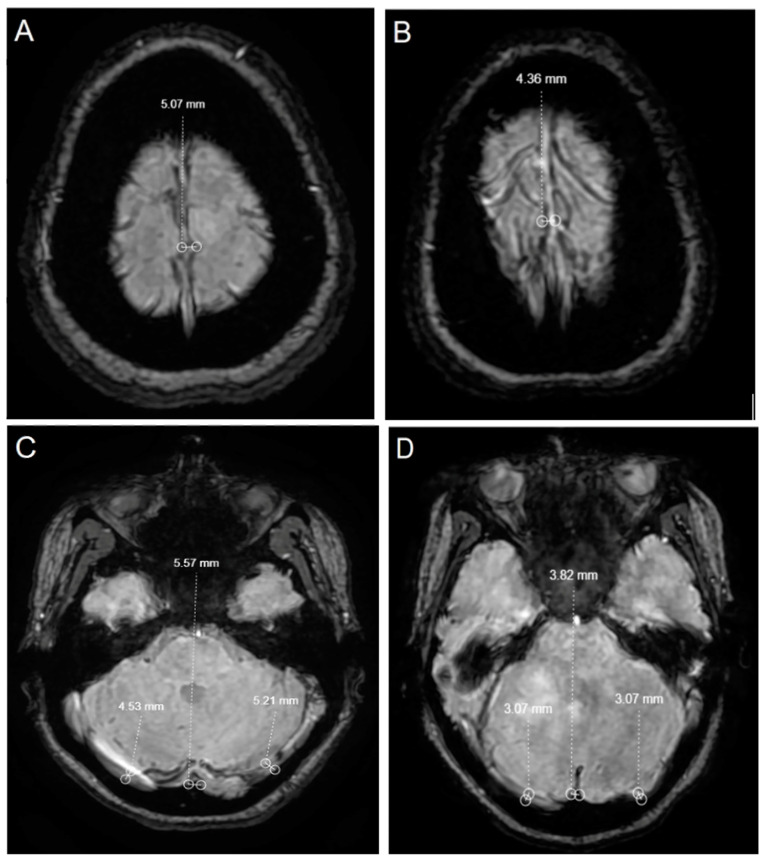
Axial SWAN images showing the superior sagittal sinus (**A**,**B**), right and left transverse sinuses, and sinus rectus measurements (**C**,**D**) of a 44-year-old female MS patient and a healthy subject.

**Table 1 diagnostics-14-01760-t001:** Clinical and demographic characteristics of patients with MS (*n* = 79).

Variables	*n* (%) or Mean ± SD
MS disease duration (years)	7.1 ± 5.4
**Clinical Form**	
PPMS	1 (1.3%)
RRMS	69 (87.3%)
SPMS	9 (11.4%)
**Used DMTs**	
None	2 (2.5%)
Alemtuzumab	2 (2.5%)
Dimethyl fumarate	17 (21.5%)
Fingolimod	24 (30.4%)
Glatiramer acetate	5 (6.3%)
Interferon beta 1a	2 (2.5%)
Natalizumab	3 (3.8%)
Ocrelizumab	16 (20.3%)
Teriflunomide	8 (10.1%)
EDSS	2.2 ± 1.8
Patients with brainstem lesion	37 (47.4%)
Patients with spinal lesion	63 (80.8%)
**CSF**	
OCB positive	59 (92.2%)
OCB negative	5 (7.8%)
IgG Index	0.9 ± 0.4
Number of OCBs	10.6 ± 5.2
Brainstem functional system score	0.2 ± 0.5
Lesion diameter (>1 cm)	62 (78.5%)
Number of lesions (>1 cm)	4.8 ± 4.6
Total number of lesions	18.5 ± 12.3
Number of gadolinium-enhancing lesions	0.5 ± 3
Number of periventricular lesions	11.2 ± 6.8
Central extinction	0.3 ± 0.7

“*n*” denotes the total number of participants in the study. The percentages are indicated in parentheses. The mean values are expressed as the means ± standard deviations (SDs). Abbreviations: MS: Multiple sclerosis; PPMS: Primary progressive MS; RRMS: Relapsing-remitting MS; SPMS: Secondary progressive MS; DMTs: Disease Modifying Therapies; EDSS: Expanded Disability Status Scale; CSF: Cerebrospinal fluid; OCB: Oligoclonal band; IgG: Immunoglobulin G.

**Table 2 diagnostics-14-01760-t002:** Comparative analysis of variables between patients with MS and healthy subjects.

Variables	Total (*n* = 146)	Healthy Control (*n* = 67)	MS (*n* = 79)	*p*-Value	*p*-Value *
Gender				0.913	
Female	116 (79.5%)	54 (80.6%)	62 (78.5%)		
Male	30 (20.5%)	13 (19.4%)	17 (21.5%)		
Age (years)	36 ± 10	34 ± 10	38 ± 10	0.011	
SSS	5.4 ± 0.9	4.8 ± 0.6	5.9 ± 0.8	<0.001	<0.001
Sinus rectus	4.6 ± 0.8	4 ± 0.7	5 ± 0.5	<0.001	<0.001
Right transverse sinus	5.2 ± 1.6	4.8 ± 1.4	5.5 ± 1.7	0.020	0.015
Left transverse sinus	4.1 ± 1.6	3.7 ± 1.4	4.4 ± 1.7	0.005	0.003

* Adjusted to Age. “*n*” denotes the total number of participants. The percentages are given in parentheses. The mean values are expressed as the means ± standard deviations (SDs). *p* < 0.05 indicated statistical significance. Abbreviations: SSS: superior sagittal sinus. The measurements are in millimeters (mm).

**Table 3 diagnostics-14-01760-t003:** Comparison of venous sinus diameters in patients with MS stratified by sex, CSF status, and the presence of brainstem and spinal lesions.

Variables	Superior Sagittal Sinus Diameter		Sinus Rectus Diameter		Right Transverse Sinus Diameter		Left Transverse Sinus Diameter	
	Mean ± SD	*p*-Value	Mean ± SD	*p*-Value	Mean ± SD	*p*-Value	Mean ± SD	*p*-Value
**Gender**		0.574		0.169		0.431		0.960
Female	5.9 ± 0.8		5 ± 0.5		5.6 ± 1.6		4.5 ± 1.6	
Male	6 ± 0.8		5.2 ± 0.6		5.2 ± 2		4.5 ± 1.5	
**OCB**		0.839		0.798		0.680		0.434
Positive	5.9 ± 0.8		5.1 ± 0.6		5.6 ± 1.8		4.5 ± 1.6	
Negative	6 ± 0.9		5.1 ± 0.3		5.2 ± 2.2		3.8 ± 1.9	
**Brainstem lesion**		0.583		0.497		0.323		0.054
Present	5.9 ± 0.8		5.1 ± 0.6		5.7 ± 1.8		4.9 ± 1.7	
Absent	5.8 ± 0.8		5 ± 0.5		5.3 ± 1.6		4.2 ± 1.4	
**Spinal lesion**		0.010		0.021		0.441		0.255
Present	6 ± 0.8		5.1 ± 0.5		5.6 ± 1.7		4.6 ± 1.5	
Absent	5.4 ± 0.5		4.7 ± 0.5		5.2 ± 1.8		4.1 ± 1.9	

*p* values represent the statistical significance of the differences between groups. The measurements are given in millimeters (mm). Abbreviations: MS; multiple sclerosis, OCB; oligoclonal band, mean ± SD; mean ± standard deviation.

**Table 4 diagnostics-14-01760-t004:** Correlation analysis between various variables and venous sinus diameter in patients with MS.

Variables	SSS Diameter	Sinus Rectus Diameter	Right Transverse Sinus Diameter	Left Transverse Sinus Diameter
	*r*/*p*	*r*/*p*	*r*/*p*	*r*/*p*
EDSS	−0.006/0.959	−0.122/0.323	0.025/0.841	−0.046/0.708
Age (years)	−0.100/0.415	−0.133/0.275	−0.270/0.025	−0.146/0.233
IgG index	0.106/0.665	0.301/0.211	−0.062/0.799	0.059/0.811
Brainstem functional system score	−0.126/0.306	−0.154/0.210	0.092/0.454	0.139/0.259
Number of OCBs	−0.058/0.874	0.203/0.574	0.145/0.688	0.220/0.541
Total number of lesions	0.008/0.949	−0.079/0.520	−0.041/0.737	0.158/0.195
Number of GD enhancing lesions	0.075/0.538	0.039/0.752	−0.063/0.607	−0.008/0.946
Number of periventricular lesions	0.108/0.398	0.008/0.953	0.118/0.358	0.125/0.327
Central extinction	0.189/0.139	−0.140/0.275	−0.070/0.588	−0.154/0.229

“*r*” indicates the correlation coefficient, which measures the strength and direction of a linear relationship between two variables. *p* < 0.05 indicates a significant correlation. Measurements involve various venous sinus diameters in millimeters (mm). Abbreviations: MS: Multiple sclerosis; EDSS: Expanded Disability Status Scale; OCB: Oligoclonal band; IgG: Immunoglobulin G; GD: gadolinium.

## Data Availability

The datasets used and/or analyzed during the current study are available from the corresponding author upon reasonable request.

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
