# Peer review of "Differential Analysis of Venous Sinus Diameters: Unveiling Vascular Alterations in Patients with Multiple Sclerosis"

_diagnostics, 2024, doi:10.3390/diagnostics14161760_

Round 1

Reviewer 1 Report

Comments and Suggestions for Authors

There are major methodology concerns:

How was the matching method? In methods it is said that HC and MS are matched, however, in results section it is stated that they are different in age. 

How was the sampling method? Was it randomly? It seems that this part is biased in this study. 

The only correlation was seen between diameters and age. How was these correlations in HCs? Since these two groups were not matched for age, possibly the age correlation can be a biased finding. 

The conclusion is not adding anything to the available facts. What this study adds? 

There should be linear/logistic regression models, adjusted for at least age. The age seems to be an important covariate that has not been considered sufficiently in statistics. 

Reviewer 2 Report

Comments and Suggestions for Authors

Methods

Statistical analyses are appropriate in this cross-sectional study. However: 

1) You stated that this is an age and gender matched control study. However, age sounds statistically significant between groups (Table 1). The exclusion process may have contributed to that. This point should be discussed as limitation point.

2)MRI protocol, in this section all the MRI parameters should be clarified. For instance, you reported the “central extinction” in table 2 although it is not clarified before.  

Results

1)As first, I suggest to report the featured of MS patient so table 2 should became table 1. 

2)Authors carried out an analysis on RRMS patients (n=69/79) as reported also in table 3. Results from this analysis are almost consistent to those conducted on 79MS patients vs control. Since the RRMS patient represented the 87.3%, the main analysis was driven by these patients and the secondary analysis (on RRMS) should be suppressed.  All the analyses restricted to RRMS patients may be avoided due to the limited size of the cohort.  

Rationale and discussion 

In the introduction section you explained the rational of this study that is to explore whether patients with MS have venous system abnormalities compared to controls. Indeed, you say that the vascular abnormalities, affecting the CSF circulation, may have a role in MS pathophysiology. 

The involvement of the venous system in MS is not recently. Indeed, a typical pathognomonic MRI sign is the central vein sign reflecting the perivenular localization of typical MS lesions. 

According to recent studies, it would seem that MS progression is linked to “subpial surface in” and “ependymal in” gradients causing leptomeningeal and cortical damage and deep gray matter damage, respectively (Magliozzi R, Fadda G, Brown RA, Bar-Or A, Howell OW, Hametner S, Marastoni D, Poli A, Nicholas R, Calabrese M, Monaco S, Reynolds R. "Ependymal-in" Gradient of Thalamic Damage in Progressive Multiple Sclerosis. Ann Neurol. 2022 Oct;92(4):670-685. doi: 10.1002/ana.26448. Epub 2022 Jul 30. PMID: 35748636; PMCID: PMC9796378.)   

You found that “the number of perivenular lesions was significantly higher in patients with a disease duration of more than 10 years (p=0.032), a finding also applicable to RRMS patients”

This is not surprising since perivascular infiltrates are associated with early demyelinating activity in MS and recently also to MS progression (Nicholas R, Magliozzi R, Marastoni D, Howell O, Roncaroli F, Muraro P, Reynolds R, Friede T. High Levels of Perivascular Inflammation and Active Demyelinating Lesions at Time of Death Associated with Rapidly Progressive Multiple Sclerosis Disease Course: A Retrospective Postmortem Cohort Study. Ann Neurol. 2024 Apr;95(4):706-719. doi: 10.1002/ana.26870. Epub 2024 Jan 12. PMID: 38149648.)

Thus, the perivascular inflammation seems to be predominant in MS and your result may be reviewed in this view. Indeed, the venous sinus alterations may be the consequence of intrathecal inflammation with the accumulation of lymphoid-like infiltrates, B lymphocytes and other.  Indeed, the large parenchymal veins are found close to the ependymal surface and drain toward the periventricular surface. Also, perivascular spaces have been shown to increase in size and number in MS brain.

Thus, the abnormalities of the venous system in patients with MS probably is an epiphenomenon of intrathecal inflammation and it is not a primum movens of the disease. I suggest to rearrange the article according these evidence.

Kind regards.

Reviewer 3 Report

Comments and Suggestions for Authors

The authors reported a manuscript about the Differential Analysis of Venous Sinus Diameters: Unveiling Vascular Alterations in Multiple Sclerosis. 

After careful review, I have some suggestions:

- in the abstract section: the background should include the knowledge about the topic and after the paper's aims.

- any abbreviation form needs a full form;

- English needs a grammar and syntax review;

- the authors should improve the section discussion. how do they explain this result? some hypothesis?  

- some imageas of cases coul improve a quality of paper (normal group VS MS group)

Comments on the Quality of English Language

need revision

- English needs a grammar and syntax review;

Round 2

Reviewer 1 Report

Comments and Suggestions for Authors

'The control group consisted of 67 age and healthy individuals recruited from general neurology outpatient clinics. ' This sentence does not make sense. 

The paper needs comprehensive English revision. 

Comments on the Quality of English Language

 The paper needs comprehensive English revision. 

Author Response

English editing was done throughout the article with the support of Cureus.

Reviewer 2 Report

Comments and Suggestions for Authors

All the issues have been clarified. Now the manuscript can be published.

Author Response

Thank you very much for your contribution to the article.

Reviewer 3 Report

Comments and Suggestions for Authors

Good improvement in any section

Comments on the Quality of English Language

Fine

Author Response

(The authors gave the same response as above.)
